# Physicochemical and Biological Effects on Activated Sludge Performance and Activity Recovery of Damaged Sludge by Exposure to CeO_2_ Nanoparticles in Sequencing Batch Reactors

**DOI:** 10.3390/ijerph16204029

**Published:** 2019-10-21

**Authors:** Qian Feng, Yaqing Sun, Yang Wu, Zhaoxia Xue, Jingyang Luo, Fang Fang, Chao Li, Jiashun Cao

**Affiliations:** 1Key Laboratory of Integrated Regulation and Resource Development on Shallow Lakes, Ministry of Education, Hohai University, Nanjing 210098, China; xzx223@sina.com (Z.X.); luojy2016@hhu.edu.cn (J.L.); ffang65@hhu.edu.cn (F.F.); lichao0609@163.com (C.L.); caojiashun@163.com (J.C.); 2College of Environment, Hohai University, Nanjing 210098, China; syqhhu@163.com (Y.S.); wuyang1026@hhu.edu.cn (Y.W.)

**Keywords:** cerium oxide nanoparticles, key enzymes, extracellular polymeric substances, microbial community, activity recovery of damaged sludge

## Abstract

Recently, the growing release of CeO_2_ nanoparticles (CeO_2_ NPs) into sewage systems has attracted great concern. Several studies have extensively explored CeO_2_ NPs’ potential adverse impacts on wastewater treatment plants; however, the impaired activated sludge recovery potentials have seldom been addressed to date. To explore the physicochemical and biological effects on the activated sludge performance and activity recovery of damaged sludge by exposure to CeO_2_ NPs in sequencing batch reactors (SBRs), four reactors and multiple indicators including water quality, key enzymes, microbial metabolites, the microbial community structure and toxicity were used. Results showed that 10-week exposure to higher CeO_2_ NP concentration (1, 10 mg/L) resulted in a sharp decrease in nitrogen and phosphorus removal efficiencies, which were consistent with the tendencies of key enzymes. Meanwhile, CeO_2_ NPs at concentrations of 0.1, 1, and 10 mg/L decreased the secretion of tightly bound extracellular polymeric substances to 0.13%, 3.14%, and 28.60%, respectively, compared to the control. In addition, two-week recovery period assays revealed that the functional bacteria *Proteobacteria*, *Nitrospirae* and *Planctomycetes* recovered slightly at the phyla level, as analyzed through high-throughput sequencing, which was consistent with the small amount of improvement of the effluent performance of the system. This reflected the small possibility of the activity recovery of damaged sludge.

## 1. Introduction

Cerium oxide nanoparticles (CeO_2_ NPs), as metallic/metal oxide nanoparticles, have been widely exploited because of their nanometer size and specific properties [1]. In terms of the domains of industry, agriculture and commerce, CeO_2_ NPs have been extensively used in UV absorbers and antioxidants [2], fuel additives [3], gas sensors and ion membranes [4,5]. However, due to their specific physicochemical properties, the negative effects of CeO_2_ NPs can no longer be neglected, especially regarding their life cycle in nature [6]. CeO_2_ NPs have been shown to produce toxicity in microalgae by oxidative stress [7]. Nanocrystalline CeO_2_-based catalysts have been proved to be harmful to fish and daphnids [8]. Meanwhile, it has been demonstrated that the inhalation of CeO_2_ NPs can induce pulmonary inflammation in rats [9] and also poses a potential hazard to the human body [10].

With the rapid application and disposal of CeO_2_ NP products, the release of CeO_2_ NPs into sewage collection systems and wastewater treatment plants (WWTPs) is becoming a growing concern [11]. As the last barrier before entering the environment, WWTPs are considered to remove the majority of nanoparticles via a series of physical, chemical and biological processes, along with a considerable risk of exposure [12]. Therefore, it is significant to explore the potential effects of CeO_2_ NPs on biological wastewater treatment systems based on various reaction systems. To date, a number of studies have paid close attention to biological phosphorus removal [13] and the physicochemical characteristics of extracellular polymeric substance (EPS) fractions [14] in biofilms in a sequencing batch biofilm reactor (SBBR). Protective mechanisms of the response of EPS to chronic CeO_2_ NP exposure were also taken into account [15], as well as the sludge dewatering performance in SBBR [16]. In addition to the biofilm system in SBBR, the activated sludge system was also widely discussed, especially regarding the performance of the system, microbial enzymatic activity [17] and bacterial community shifts [18] in a sequencing batch reactor (SBR). Nevertheless, recent research has been limited only to the system performance in the presence of CeO_2_ NP exposure [11,14]. Simultaneously, little is known regarding how the activated sludge will perform when freed from the chronic exposure of CeO_2_ nanoparticles and the response of the microbial community structure, which is extremely vital to the stability of the activated sludge system. The potential recoverability of CeO_2_ NP-damaged sludge in nitrogen and phosphorus removal systems has seldom been addressed to date. Limited research has indicated the metabolic recovery capacities of impaired microorganisms under nanoparticle exposure. Recently, only Wu et al. [19] have discussed the possibility of microbial activity recovery under ZnO NP stress when exploring the effects of dissolved oxygen on *Nitrosomonas europaea*.

Therefore, more information is needed to close the gap in the current knowledge regarding the presence and absence of CeO_2_ NP exposure, especially focused on the process of the activated sludge system. The major purposes of this study were as follows: (1) to further study the characteristics of sludge during chronic CeO_2_ NP exposure in SBRs; (2) to evaluate the stability of the system by comparing the indicators of nutrient removal efficiency, key enzyme activity, microbial metabolite ability, and toxicity during the beginning and the end exposure period; and (3) to explore how the microbial community will respond during the recovery period and determine its recovery ability.

## 2. Materials and Methods

### 2.1. Experimental Set-Up

The activated sludge (AS) was cultured in four SBRs of a working volume of 4 L with a concentration of 3.0 g/L biomass. AS was taken from the secondary sedimentation tank at Jiangning Development Zone Sewage Treatment Plant in Nanjing, China. The SBRs were operated daily in 8 h cycles, including a 10 min feeding phase, 2 h anaerobic phase, 4 h aerobic phase, 1 h setting phase, 10 min draining phase, and a 40 min idle phase. The reactors were fed with synthetic water, which was composed of chemical oxygen demand (COD), 300 mg/L; NH_4_Cl, 120 mg/L; KH_2_PO_4_, 24 mg/L; MgSO_4_·7H_2_O, 20 mg/L; and CaCl_2_·5H_2_O. Micronutrient compositions were as follows: H_3_BO_3_, 0.1 g/L; FeSO_4_·7H_2_O, 6 g/L; CuSO_4_·5H_2_O, 0.08 g/L; MnCl_2_·4H_2_O, 0.08 g/L; Na_2_MoO_4_·2H_2_O, 0.06 g/L; ZnSO_4_·7H_2_O, 0.12g/L; CoCl_2_·6H_2_O, 0.1g/L; and NiCl_2_·6H_2_O, 0.1 g/L. Each reactor was bubbled with a volume exchange ratio of 50% through a fine-bubble aerator connected to their bottom. The parameters of the reactor were controlled as follows: a dissolved oxygen (DO) concentration of 2 mg/L, a temperature of 20 ± 2 °C, and a pH value of 7.0–8.0. The hydraulic retention time (HRT) and sludge retention time (SRT) were 8 h and 14 days, respectively.

### 2.2. CeO_2_ NPs Treatments

The CeO_2_ NPs (100 mg), purchased from Shanghai Jingchun Scientifical Co. Ltd., were first placed in ultrapure water (1.0 L, pH 6.9) for the ultrasonic treatment (25 °C, 1 h, 250 W, 40 kHz). The mean particle size of the CeO_2_ NP suspension was 70–150 nm after the analysis by dynamic light scattering (Malvern Instruments Co. Ltd., Malvern, UK). A volume of 0.1 mg/L CeO_2_ NPs was chosen as the environmentally relevant concentration, with 10 mg/L as the upper concentration limit in the article, and 1 mg/L was chosen as the transition value.

### 2.3. CeO_2_ NP Exposure and Recovery Experiment

The entire running process was composed of a start-up period, a period of intermediate exposure to CeO_2_ NP, and an exposure relief period. The SBRs operated continuously for 12 weeks after the system remained stable, including 10 weeks with CeO_2_ NP exposure to study the effect of NPs on the activity of active sludge, and two weeks without CeO_2_ NP exposure to study the recovery ability of damaged sludge in the first 10 weeks. After the 10-week exposure period, the activated sludge in SBRs was washed out three times with deionized water to remove the residual CeO_2_ NPs in the reactors, and then a two-week recovery experiment was carried out. At the same time, the total CeO_2_ and dissolved CeO_2_ contents were monitored to ensure the NP removal efficiency. After three times of elution, the concentration of total Ce in the supernatant was less than 0.01 mg/L, which indicated that the exposure of CeO_2_ NPs was relieved to some extent.

### 2.4. Analytical Methods

During the CeO_2_ nanoparticle loading process, the wastewater effluent samples of the SBRs were collected for ammonia nitrogen (NH_4_^+^-N), nitrate nitrogen (NO_3_^−^-N), nitrite nitrogen (NO_2_^−^-N), total phosphorus (TP) and chemical oxygen demand (COD) analysis. NH_4_^+^-N, NO_3_^−^-N, NO_2_^−^-N, TP, biomass concentration and COD were measured in duplicate by the standard methods [20].

Furthermore, the morphology of the activated sludge was observed by scanning electron microscope SEM (JEOLJEM-1400, 120kV, Tokyo, Japan). The soluble microbial products (SMP), loosely bound extracellular polymeric substances (LB-EPS), and tightly bound extracellular polymeric substances (TB-EPS) were extracted separately, as well as the polysaccharides, proteins, and humic acid content by Hou et al. [21] and Wang et al. [15]. The lactate dehydrogenase (LDH) release assay was measured with an LDH kit (Jiancheng Bioengineering Co. Ltd., Nanjing, China) [11]. Furthermore, the detection of the microbial diversity could be carried out using high-throughput sequencing technology for microbial analysis. In addition, measurements of ammonia monooxygenase (AMO), nitrite oxidereductase (NOR), nitrate reductase (NAR) and nitrite reductase (NIR) activity were performed according to the method of Zheng et al. [22]. The microbial community structure was analyzed by miSeq high-throughput sequencing according to the method of Li et al. [23]. Triplicate samples were collected from each SBR reactor to ensure the integrity of the AS samples, and all results were presented on average.

### 2.5. Energy Dispersive Spectroscopy Analysis

The mixed solution (40 mL) was first dispersed into four centrifuge tubes (each 10 mL). These four centrifuge tubes were then centrifuged at 3000 rpm/min for 5 min. After pouring the supernatant, the lotion (1.0 mmol/L EDTA, 0.1 mol/L NaCl) was added to the centrifuge tubes to restore the original volume (10 mL). Then, the mixtures were transferred to the oscillating tube (50 mL) and oscillated in a constant temperature oscillator at 150 rpm/min for 30 min to remove CeO_2_ NPs adsorbed on the surface of the sludge [24]. After repeating the steps above, the centrifuge tubes were centrifuged at 3000 rpm/min for 5 min again. Then, we poured out the supernatant and dried the remaining sludge at 60 °C. Finally, the dried sludge was ground into powder for the spectrum analysis, which could be used to determine the proportion of intracellular cerium.

### 2.6. Data Analysis

All assays were conducted in triplicate. For statistical analysis, the experimental values were compared to their corresponding control values. An analysis of variance (ANOVA) was used to test the significance of the results, and *p* < 0.05 was considered to be statistically significant, as similarly described by Cao et al. [25].

## 3. Results and Discussion

### 3.1. Sludge Properties under Exposure to CeO_2_ NPs

Figure 1 clearly showed the size distribution of sludge under CeO_2_ NP stress. The particle size distribution ranges concentrated between 0–1.5 mm were similar in the exposure period to different CeO_2_ NP concentrations. The sludge size increased from 0.274 mm to 0.495 mm in the control reactor, indicating the sludge was granulated. It can be seen that the sludge particle size had increased under the long-term effect of CeO_2_ NPs, but remained smaller than the average particle size of the control, in the comparison in Figure 1a,b. This can probably be attributed to the presence of nanoparticles, which can activate sludge by stress and inhibit the growth of particle size. Meanwhile, dissolved nanoparticles tended to be adsorbed on the surface of the activated sludge, making it expand, which could be observed evidently in reactor 1 (R1) at the concentration of 0.1 mg/L. This can be interpreted as the presence of high-concentration nanomaterials easily being able to induce self-aggregation and thus reducing the positive contact with sludge.

The magnified SEM images of the activated sludge from R4 are given in Appendix A. The yellow circles in Appendix A represent the filamentous bacteria, spherical bacteria, rod-shaped bacteria and the attached CeO_2_ NPs, respectively. It can be seen from Appendix A that the activated sludge in R4 was primarily composed of spherical and rod-shaped bacteria. With the increase of the concentration of CeO_2_ NPs, increasing numbers of CeO_2_ NPs were absorbed onto the surface of the spherical and rod-shaped bacteria. However, the numbers of spherical and rod-shaped bacteria were significantly decreased. A number of studies showed that most phosphorus-accumulating bacteria are spherical bacteria, and most of the denitrifying bacteria are rod-shaped bacteria [26]. These bacteria might be deactivated when they contact with CeO_2_ NPs for a long time. This was confirmed by the decreased nitrogen removal efficiency under the higher concentration of CeO_2_ NPs (10 mg/L), as described below in Section 3.2. Furthermore, Appendix A shows that a certain amount of CeO_2_ NPs were absorbed onto the surface of the activated sludge.

### 3.2. Nutrient Removal and Related Key Enzymes during CeO_2_ NP Exposure

The response of reactor stability to nano-CeO_2_ stimulation can be visually detected by effluent water quality. COD is a primary factor indicating the organic removal stability in wastewater treatment. Almost all COD values (>90%) in SBRs were observed to be removed, and the marginal difference was presented among four reactors when setting up, respectively. After 10-week exposure, the COD removal efficiency at a CeO_2_ NP concentration of 10 mg/L was 73.33%, which was much lower than the values of 82.00% for 1 mg/L and 86.67% for 0.1 mg/L. Thus, a higher concentration of CeO_2_ NPs had a negative influence on the activated sludge according to our results (Figure 2), which was consistent with the conclusion of Wang et al. [17]. It can be interpreted that COD removal mainly depended on the heterotrophic bacteria in activated sludge, meaning that the decrease of COD removal was acknowledged as the inhibition of the activity of the heterotrophic bacterial response to exposure.

At the initial stage of CeO_2_ NP exposure, the ammonia nitrogen removal efficiency in R2, R3 and R4 systems was about 96–98%, which was almost the same as the control (Figure 3). However, after 10-week exposure to CeO_2_ NPs, the NH_4_^+^-N removal efficiencies in the four reactors were 98.78%, 98.26%, 96.85% and 85.33%, respectively. This indicated that the inhibition of NH_4_^+^-N removal efficiency increased with the increase of CeO_2_ NP exposure time and concentration. It can be inferred that when the CeO_2_ NP exposure time to the activated sludge system and concentration exceed a certain limit, it will inevitably affect the activity of related enzymes and the pathway of material transformation in the nitrification process. Hence, this requires the risk assessment of hazardous material leakage to be measured on a longer time scale. NO_3_^−^-N and NO_2_^−^-N accumulated mainly in the aerobic stage, after which the NO_2_^−^-N content increased gradually with the oxidation of NH_4_^+^-N under the action of nitrite bacteria. The NO_2_^−^-N content reached peaks of 4.29, 3.92, 3.73 and 2.36 mg/L, respectively, after 240 min during the end of the exposure period (Figure 3Bb). After that, NO_2_^−^-N was gradually oxidized to NO_3_^−^-N by nitrifying bacteria. Figure 3 also shows that the phosphorus removal efficiency was inhibited after 10-week exposure, although several studies declared that the bacteria responsible for phosphorus removal are less sensitive than the nitrogen removal-related bacteria [27].

In order to further explore the mechanism of the effects of CeO_2_ NP exposure on microbial performance in SBRs, the effects of different concentrations of CeO_2_ NPs on the activity of key enzymes related to the nitrogen and phosphorus removal process are displayed in Figure 4. It can be seen from Figure 4a that the relative activity of AMO was inhibited under the long-term high concentration of CeO_2_ NPs exposure, which would lead to the reduction of NH_4_^+^-N conversion rate. However, from Figure 4c, the relative activity of NOR was slightly higher than that of the control group, indicating that the transformation of NO_2_^-^-N to NO_3_^-^-N was not hindered. These two effects together led to the reduction of nitrite accumulation. In general, the results of the enzyme activity measurement were consistent with the tendency of nitrogen and phosphorus removal shown in Figure 3. CeO_2_ NPs were vital in the activity of key enzymes related to nitrogen removal. It was reported that AMO and NOR played an important role in nitrification, and denitrification was related to NAR and NIR [28,29]. The relative activity of NIR was 79.00% under 10-week exposure and 89.00% under 1-day exposure, together with the 70.00% and 87.00% of relative activity of NAR under the same conditions, respectively (Figure 4), in which the variation tendency coincided with Figure 3. Compared with NAR and NIR, the activities of AMO and NOR did not change significantly, which may indicate that NAR and NIR were more sensitive to CeO_2_ NPs on a longer time scale.

### 3.3. Response of Microbial Metabolites to CeO_2_ NP Exposure and Toxicity Analysis

In order to resist the toxicity of nanomaterials, microorganisms tend to promote the production of LB-EPS. Due to excessive LB-EPS secretion, activated sludge flocs became unstable after exposure to CeO_2_ NPs, which was consistent with the variation of sludge size in Figure 1. LB-EPS in the outer layer played a key role in resisting CeO_2_ NPs because of its loose and porous structure, which provided a large number of sites for nanoparticles [14,30]. The content of LB-EPS increased from 21.91 mg COD/gVSS in R1 to 22.48 mg COD/gVSS in R4 under 1-day exposure to CeO_2_ NPs (Figure 5). This indicated that more LB-EPS was secreted by microorganisms to resist the toxicity of CeO_2_ NPs, while chronic exposure to CeO_2_ NPs inhibited the activity of microorganisms, resulting in the decrease of LB-EPS.

TB-EPS had a compact structure, which prevented nanomaterials from entering cells and protected microorganisms from toxicity [14]. Therefore, the content of TB-EPS increased with the CeO_2_ NP accumulation during 1-day exposure, which also proved that the variation of nitrogen and phosphorus removal efficiency was not significant in the presence of 1-day exposure to CeO_2_ NPs (Figure 3). The accumulation of TB-EPS decreased the risk to some extent [31]; however, a decrease of TB-EPS was observed not only in polysaccharides and proteins, but also in humic acids under long-term exposure, and the higher the exposure concentration, the greater the decrease. The content of TB-EPS exposed to the concentration of 0.1 mg/L was 46.65 mg COD/gVSS, which was 2.37% lower than the control, while the content of TB-EPS under a 10 mg/L exposure concentration decreased to 40.90%. This illustrated that the toxicity of nanoparticles can inhibit microorganisms when the time scale is taken into account. At the end of the 10-week exposure, the content of LB-EPS in R2, R3 and R4 decreased by 0.13%, 3.14% and 28.60%, respectively, compared with that in R1 at 21.99 mg COD/gVSS. Therefore, a conclusion can be drawn that TB-EPS was more sensitive to the external environment than LB-EPS when it changed, which was contrary to the former conclusion that LB-EPS was more rapid than TB-EPS in emergency response to the external environment [15]. This can be interpreted as the chronic CeO_2_ NP toxicity causing great damage to microbial activity, and the toxicity of the nanomaterials inside microorganisms was greater than that of the external contact.

At the same time, it has been reported that polysaccharides were beneficial to increasing the hydration size of nanomaterials [32], promoting their reunion, and preventing them from entering cells [33]. Regarding the early stage of CeO_2_ NP exposure, the increase of EPS production can reduce the toxicity of nano-particles to cells, but it may also prevent the effective transfer of the organic matter to the internal microorganisms. Proteins in EPS have enzymatic functions, which can digest and decompose macromolecules and granular substances in the cell microenvironment, thus ensuring the transfer of organic substances [14]. Compared with polysaccharides and humic acids, the protein content in LB-EPS and TB-EPS decreased significantly after chronic CeO_2_ NP exposure (Figure 5), which partly explained the relative activity of key enzymes being inhibited under long-term exposure, and the degree of inhibition was more significant with the increase of the exposure concentration, as shown in Figure 4.

The variation of SMP differed from that of EPS (Figure 5). When the time of exposure increased, the components in SMP increased significantly compared with the initial exposure period. The content of SMP exposed to 10 mg/L CeO_2_ NPs was 7.35 mg COD/gVSS, while the value in the control was 2.99 mg COD/gVSS. The content of SMP in reactors remained at about 3.00 mg COD/gVSS at the beginning of the exposure. Mei et al. [34] also found that SMP production increased with the rise of ZnO NP concentration when studying its effect on microorganisms in MBR.

Polysaccharides were more sensitive to changes in exposure concentration than humic acids in SMP (Figure 5). During 10-week exposure, protein content was gradually detected in SMP; however, the protein content in SMP was lower than the minimum detection line in the early stage of exposure, which indicated that polysaccharides and humic acids were the main components of SMP. It has been pointed out that SMP accounts for almost all COD in wastewater with good biodegradability, such as domestic and food wastewater [35]. An et al. [36] reported that the presence of a large amount of SMP would not only reduce the effluent quality of the bioreactor but also adversely affect its operation. This was consistent with the result that the COD removal efficiency decreased but SMP concentration increased under 10-week CeO_2_ NP exposure.

Proteins in SMP tend to be combined with nanoparticles to reduce their toxicity. The adsorption of EPS prevented most of the low-concentration nanoparticles from entering cells. As increasing numbers of nanoparticles became available, EPS stimulates cells to produce more EPS in order to form a thicker outer barrier to protect themselves. The combined action of EPS and SMP constitutes the mechanism of microbial cell resistance to nanoparticle toxicity. Several reports have pointed out that there is a mutual transformation between EPS and SMP [37].

As shown in Figure 5, the contents of EPS components decreased to varying degrees during long-term exposure. In addition to considering the decrease of microbial biomass and microbial activity during the exposure, EPS can also be hydrolyzed into SMP [35], which was consistent with the increasing content of SMP discovered at the end of exposure. Therefore, the increase of humic acid and polysaccharide in SMP can be attributed to the hydrolysis of EPS stimulated by CeO_2_ NPs. A low concentration of humic acid can stabilize the suspension of nanoparticles, while a high concentration can induce aggregation. When the concentration of humic acid reached the threshold, the toxicity of CeO_2_ NPs decreased with the increase of its size. In addition, some CeO_2_ NPs can be adsorbed by polypeptides in protein to reach a stable state, which can also reduce the diffusion of CeO_2_ NPs into cells to a certain extent. The increase of SMP production is another way for microorganisms to cope with the toxicity of nanoparticles besides the increase of EPS secretion.

LDH is one of the important indicators of energy metabolism. LDH content is usually used to indicate cell integrity and evaluate the effect of toxicants on cell growth and survival [22]. Appendix A demonstrated that the relative amount of LDH released during exposure tended to increase, indicating the damage of the cell membrane and the decrease of cell integrity in the presence of long-term CeO_2_ NPs. At the initial stage of exposure, the highest LDH release at the concentration of 10 mg/L was 110.00% compared to the control group; however, the LDH release under a low concentration (<1 mg/L) showed almost no difference to that of the control, at 101.00% for 0.1 mg/L and 105.00% for 1 mg/L.

This indicated that the cell damage was lower at the early stage of exposure, which was consistent with the data shown in Figure 4 that the short-term effect of CeO_2_ NPs on key enzyme activities was reduced. More LDH was released at the end of exposure, indicating more cell damage. Nanoparticles can be adsorbed by EPS secreted by microorganisms. The encapsulation of EPS on cell membranes prevented CeO_2_ NPs from entering into cells from the outside, and nanoparticles could also enter cells through cell membranes, resulting in protein denaturation and deoxyribonucleic acid (DNA) damage, which resulted in the damage of key enzymes involved in nitrogen and phosphorus removal, consistent with Figure 4. At the end of CeO_2_ NP exposure, a higher LDH release corresponded to the stronger inhibition of key enzymes. With the increase of exposure concentration, the release of LDH increased, and the inhibition of key enzymes became more obvious.

### 3.4. System Stability Assessment When Relieved from CeO_2_ NP Exposure

Freed from long-term exposure to sludge containing large amounts of CeO_2_ NPs, the reactors continued to operate. It was observed that the performance of activated sludge in the reactors did not recover significantly within the two-week recovery time after being relieved from the 10-week exposure to different CeO_2_ NP concentrations. In R4, the removal efficiencies of nitrogen and phosphorus were stable at 86.12% and 45.26%, respectively, which were only 1.09% and 4.63% higher than those during the period of exposure. It is likely that, during the two-week activity recovery of damaged sludge, CeO_2_ NPs in the form of adhesion were not completely removed by sludge discharge, leaving residual NPs. In addition, CeO_2_ NPs entering activated sludge microbial cells still played an important role in inhibiting microbial biological activity. On the other hand, an energy spectrum analysis of microbial cells after the two-week recovery of damaged sludge was carried out. Appendix A shows that the content of cerium in R2 with 0.1 mg/L of CeO_2_ NPs was 0.30%, while in R3 and R4, the contents were 0.50% and 0.70%, respectively. Previous studies have shown that nanoparticles can easily enter cells and bind to cells [38,39]; however, under the long-term CeO_2_ NP stress, the key enzymes playing a decisive role in wastewater treatment had not increased, which was consistent with the above-mentioned conclusion that the effect of activated sludge on wastewater treatment remained at a low level. Further, the subsequent analysis of the microbial community structure also confirmed the difficulty of recovery of damaged sludge.

### 3.5. Shifts of the Microbial Community When Relieved from CeO_2_ NP Exposure

Microorganisms, as decomposers in activated sludge, play an important role in material circulation and energy transfer, are the cornerstone of maintaining the stability of the system. High-throughput sequencing was used to analyze the shift of the microbial community structure in four reactors to explore the possibility of microbial community recovery, especially focusing on the damaged sludge within the two-week recovery time after relief from the 10-week exposure to different CeO_2_ NP concentrations. The Alpha diversity statistics of the four reactors are shown in Table 1; here, the higher the Shannon index and ACE/Chao1 index, the higher the diversity of microbial community in this study. Despite a two-week recovery period, the richness and diversity of the microbial community in damaged sludge exposed to different concentrations of CeO_2_ NPs were still below the control. This indicated that the addition of nanoparticles had a negative impact on the system. Although several reports pointed out that microorganisms in the activated sludge system displayed self-adjustment and adaptability, and could maintain the stability of the system and resist the adverse environment by increasing biodiversity, it is worthy of note that the self-repairing ability of microorganisms was limited when the harm reached a certain limit. As can be seen from Table 1, the microbial diversity and richness in reactors that had been exposed to different CeO_2_ NP concentrations were almost below the control, and the higher the exposure concentration, the greater the gap. Although this result was inconsistent with that Wang et al. [15], who pointed out that 0.1 mg/L of CeO_2_ NP treatment exhibited more detrimental effects than 10 mg/L, this can be explained by the biofilm being better than floc sludge under stressing external pollutants due to its compact structure.

In order to further explore the shift of the microbial community in damaged sludge during the recovery period of two weeks, the changes of microbial communities at the phyla level in the four reactors are clearly shown in Figure 6. *Proteobacteria* accounted for the highest proportion of the phyla level that had been classified (more than 1%) and comprised more than 55% of the eight samples, followed by *Bacteroidetes*, *Firmicutes* and *planctomycetes*, with proportions of 12.78–18.45%, 1.04–7.62%, and 0.76–6.36% respectively (Figure 6). *Nitrospirae*, *Acidobacteria*, *Chloroflexi* and other common bacteria were also present in SBRs, which all played an important role in sewage treatment. The proportions of dominant bacteria in each sample were different. In the four samples at the end of 10 weeks of CeO_2_ NPs exposure, the richness of *Acidobacteria*, *Chloroflexi*, *Planctomycetes* and *Actinobacteria* decreased in varying degrees with the increase of exposure concentration. This indicated that the exposure of CeO_2_ NPs had a negative effect on the microbial community, and the higher the exposure concentration, the more significant the inhibition. Thus, even in the two-week recovery period after removing CeO_2_ NP exposure, the microbial community did not recover significantly. *Proteobacteria*, *Nitrospirae* and *Planctomycetes* showed a slight recovery during the recovery period. Although *Proteobacteria* contained a variety of nitrogen-fixing bacteria, and *Nitrospirae*, as a nitrite bacterium, can convert nitrite into nitrate [40], but even so, no significant improvement in nitrogen removal was observed during the recovery period. However, the abundances of *Actinobacteria* and *Firmicutes* [41], which were closely related to the phosphorus removal performance of activated sludge and the phosphorus uptake in biological phosphorus removal by subordinate bacteria mentioned in the literature, did not change significantly at the recovery stage compared with the end of the exposure period. Meanwhile, this partly explained why there was no significant recovery of nitrogen and phosphorus removal efficiency in the reactors exposed to CeO_2_ NPs during the recovery phase.

In order to investigate the changes of the microbial community of *Proteobacteria* during the recovery period, the heat-map of the four reactors within the two-week recovery time after relief from the 10-week exposure to different CeO_2_ NP concentrations is shown in Appendix A. It was found that the microbial diversity (from the genus level) tended to decrease with the increasing concentration of CeO_2_ NPs. *Azospira* [42], *Simplicispira* [43] and *Thauera* [44] were related to denitrification, the numbers of which increased at a low concentration of CeO_2_ NPs (0.1 mg/L). Then, with the accumulation of the action, the microbial cell membrane was destroyed, and the microorganisms died, resulting in a decrease in the number of bacteria. *Azospira* [45] had the ability to oxidize humic acid through anaerobic processes, which could use nitrate as the electron acceptor. *Simplicispira* and *Thauera* were denitrifying bacteria with the capacity of degrading organic compounds. *Dechloromonas* [46], *Thermomonas* [47] and *Planctomyces* [48] were all related to nitrogen removal, yet the number of these were reduced under an increasing concentration of CeO_2_ NPs. Moreover, *Planctomyces* could reduce nitrate. *Dechloromonas* belonged to the denitrifying phosphorus-accumulating bacteria, which were able to reduce nitrate. *Thermomonas* was related to nitrogen removal and was able to degrade phenol and nitric acid under the low-oxygen condition [49]. These changes were consistent with the performance of activated sludge in the recovery period, which also indicated that there was no significant recovery of damaged sludge activity at this stage.

## 4. Conclusions

The study attempted to investigate the physicochemical and biological effects on the activated sludge performance and activity recovery of damaged sludge under exposure to CeO_2_ NPs in SBRs. The main conclusions of the study are as follows:At the end of CeO_2_ NP exposure, the efficiency of organic matter removal and nitrogen and phosphorus removal by SBRs decreased with the increase of exposure concentration, and the inhibition phenomenon was significant.Meanwhile, the key enzymes related to nitrogen removal showed the same inhibition trend during exposure. It was found that NAR and NIR were more sensitive to the long-term toxicity of CeO_2_ NPs than AMO and NOR.In the early period of CeO_2_ NP exposure, the content of EPS increased compared with the control; however, at the end of the exposure stage, the content of EPS decreased significantly and the toxicity response of TB-EPS to CeO_2_ NPs was more sensitive than that of LB-EPS. The increase in SMP production was considered to be an autonomous way for microorganisms to cope with toxicity at the end of the period of 10-week exposure.In the two-week recovery period, it was observed in the microbial community at the phyla level that the functional bacteria *Proteobacteria*, *Nitrospirae*, and *Planctomycetes* recovered slightly during the recovery period; however, the effluent performance of the system did not recover significantly.Finally, to the best of our knowledge, the study is the first attempt to offer a systematic insight into the whole process of sludge contamination and has important guiding significance for the fate of damaged sludge based on wastewater treatment. It provides a new assessment direction for fully understanding the impact of nanoparticles on the sewage system.

## Figures and Tables

**Figure 1 ijerph-16-04029-f001:**
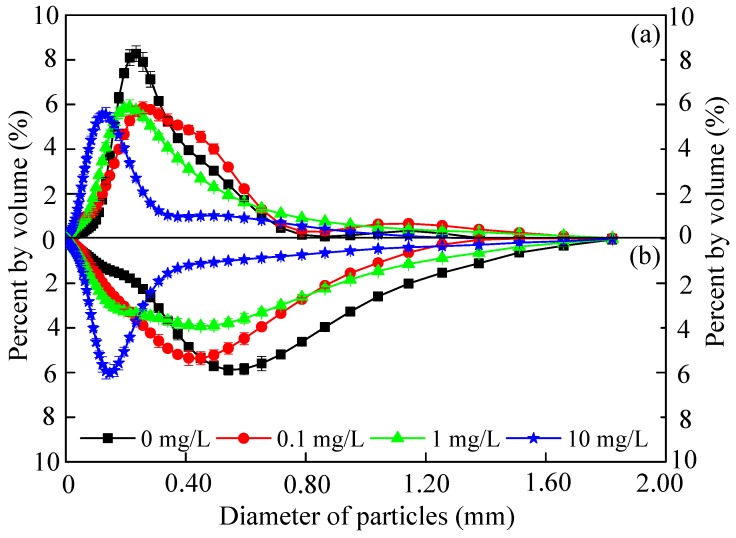
The size distribution of sludge in one cycle after 1 day (**a**) and 10 weeks (**b**) of exposure to 0 (black), 0.1 (red), 1 (green), and 10 (blue) mg/L CeO_2_ nanoparticles (NPs), respectively. Error bars represent standard deviations of triplicate measurement.

**Figure 2 ijerph-16-04029-f002:**
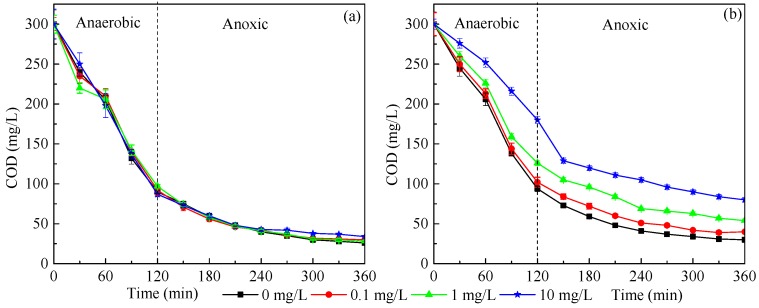
Variations of chemical oxygen demand (COD) in one cycle after 1 day (**a**) and 10 weeks (**b**) of exposure to 0 (black), 0.1 (red), 1 (green), and 10 (blue) mg/L CeO_2_ NPs, respectively. Error bars represent standard deviations of triplicate measurement.

**Figure 3 ijerph-16-04029-f003:**
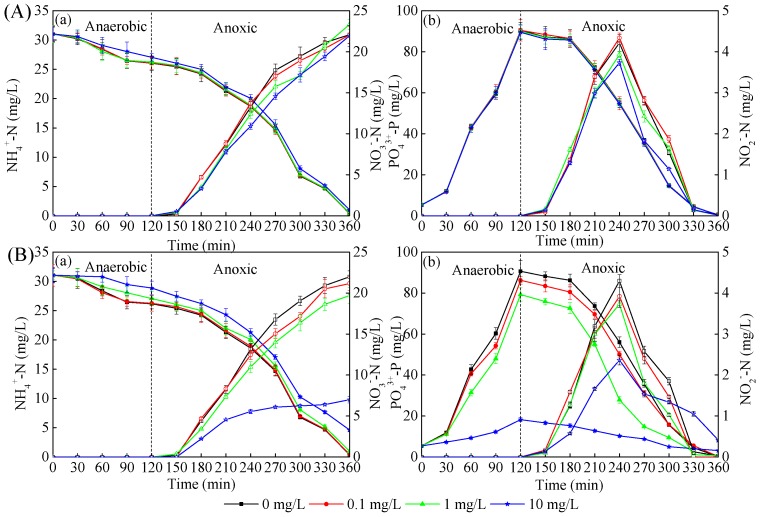
Effects of CeO_2_ NPs on the variations of (**a**) NH_4_^+^-N (full) and NO_3_^−^-N (blank), (**b**) PO_4_^3+^-P (full) and NO_2_^−^-N (blank) during one cycle after 1 day (**A**) and 10 weeks (**B**) of exposure to 0 (black), 0.1 (red), 1 (green), and 10 (blue) mg/L CeO_2_ NPs, respectively. Error bars represent standard deviations of triplicate measurement.

**Figure 4 ijerph-16-04029-f004:**
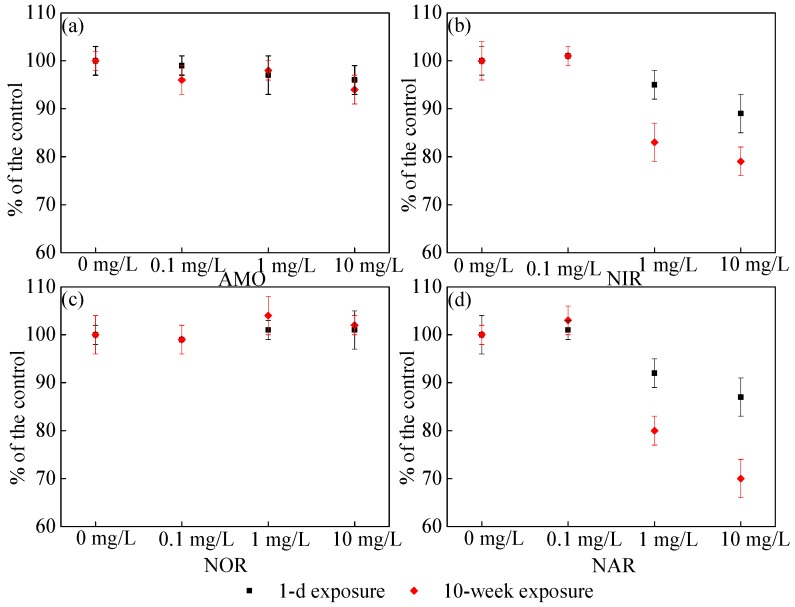
Relative activities of AMO (**a**), NIR (**b**), NOR (**c**), and NAR (**d**) in active sludge during one cycle after 1 day (■) and 10 weeks (◆) of exposure. Error bars represent standard deviations of triplicate measurement.

**Figure 5 ijerph-16-04029-f005:**
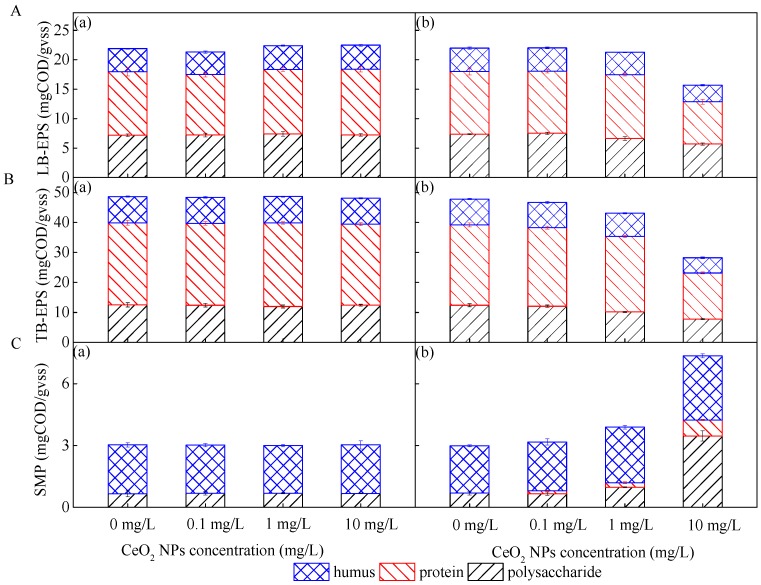
The contents and components of loosely bound extracellular polymeric substances (LB-EPS) (**A**), tightly bound extracellular polymeric substances (TB-EPS) (**B**), and soluble microbial products (SMP) (**C**) extracted from active sludge exposed to CeO_2_ NPs at different concentrations during one cycle after 1 day (**a**) and 10 weeks (**b**), respectively. Error bars represent standard deviations of triplicate measurement.

**Figure 6 ijerph-16-04029-f006:**
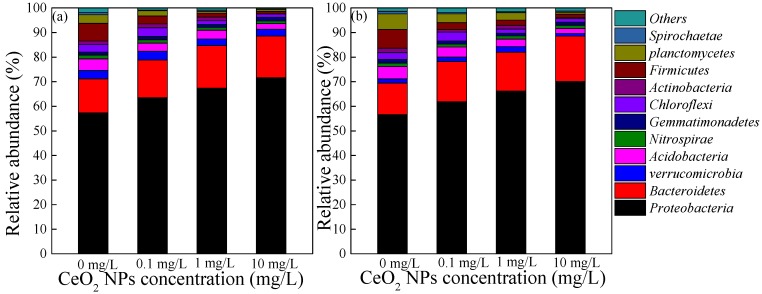
The shift of the microbial community at the phyla level in damaged sludge before (**a**) and after (**b**) the two-week recovery period after relief from different CeO_2_ NP concentrations.

**Table 1 ijerph-16-04029-t001:** The Alpha diversity of the four activated sludge reactors.

Reactor	Test	Sequence	OTU	Shannon Index	ACE Index	Chao1 Index	Coverage
R1	1	25,155	2038	5.582	7739.392	4402.294	0.991
	2	22,493	1973	5.729	7602.447	4395.162	0.992
	3	20,194	1938	5.606	7647.548	4255.880	0.993
	Average	22,614	1983	5.639	7663.129	4351.112	0.992
R2	1	20,894	1722	4.985	6704.493	3694.490	0.992
	2	23,495	2104	5.048	6602.283	3849.830	0.991
	3	22,235	1943	5.174	6669.804	3820.139	0.990
	Average	22,208	1923	5.069	6658.860	3788.153	0.991
R3	1	21,302	1985	4.805	6139.581	3695.847	0.99
	2	22,491	2039	4.923	6204.104	3593.028	0.989
	3	19,795	1598	4.708	6083.131	3539.301	0.991
	Average	21,196	1874	4.812	6142.272	3609.392	0.990
R4	1	23,021	1873	4.296	5019.492	3518.054	0.993
	2	20,493	1759	4.184	4968.105	3355.893	0.990
	3	20,320	1573	4.237	5262.933	3577.837	0.990
	Average	21,278	1735	4.239	5083.510	3483.928	0.991

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
