# Peer review of "Physicochemical and Biological Effects on Activated Sludge Performance and Activity Recovery of Damaged Sludge by Exposure to CeO2 Nanoparticles in Sequencing Batch Reactors"

_ijerph, 2019, doi:10.3390/ijerph16204029_

Round 1

Reviewer 1 Report

The work under review covers large information on the influence of CeO2 nanoparticles on the sewage sludge properties and treatment. The paper is well written, the measurements are well presented, data interpretation and application seem to have been done well. The paper adds valuable information concerning our knowledge on the biochemistry of sludge recovery processes, which is affected by the nanoparticles. I suggest the paper for publication in IJRPH, after very few minor revisions. My comments are as follows:

chapter 2.1. - please explain why this synthetic water composition was used chapter 2.2 - why the upper limit of 10 mgL CeO2 was used?

line 38 – “Meanwhile, it has..”

line 60 – please remove “sequencing batch reactors” and leave only “SBR” (this acronym is defined in line 53)

line 93 – no literature references

line 127, 128 and Fig. 1 x-axis – please replace “um” with “mm” (SI system!)

line 134 - please replace “in R1” with “in reactor 1 (R1)”

Fig. 1 y-axis – please add space “volume (%)”

Fig. 3-6 – the legends and axis description are illegible

line 227 – please explain acronym “VSS”

line 293 – please explain acronym “DHA”

line 309 – cerium

References – please complete “doi…” and correct journal title (for example “Water Research”)

Author Response

Response to Reviewer 1 Comments

Reviewer 2 Report

The manuscript investigated the physicochemical and biological effects on activated sludge performance and activity recovery of damaged sludge by exposing to CeO2 nanoparticles in sequencing batch reactors. However, lots of similar work have been studied on this topic. The novelty of this work is doubted, and I expected to see more interesting findings. In addition, the abstract section can not provide readers with more accurate information. The introduction should cover the topic’s background and the latest research progresses in a logic way. The details of the experiment procedures should be stated. Therefore, I do not recommend publishing the work in its current form in this high quality journal. My specific comments were:

Abstract is too ambiguity. Which water quality is improved? Which microbes are suppressed? Which microbes can be recovered? Lines 54-56. What is the innovation? What is the difference between present and previous research? The recent study on microbial recovery under NPs stress should be introduced in introduction. Lack of many necessary discussions. Why the nitrite accumulation reduced under 10 mg/L CeO2 NPs? Why is there no change in AMO activity, but the ammoxidation inhibition is detected? Lines 206-315. Should be further deleted and highlighted. How can high-throughput sequencing data determine the accuracy of the results without setting duplicates? Why is there only high-throughput sequencing data in the recovery phase without regular water quality indicators? There is no point in discussing the microbial diversity based on phyla level. Lines 397-400. The past literature has already systematically reported the whole process of SBR being under CeO2 stress. What is your innovation?

Author Response

Response to Reviewer 2 Comments

Point 1: Abstract is too ambiguity. Which water quality is improved? Which microbes are suppressed? Which microbes can be recovered?

Response 1: Thanks for your kind and suggestive comments. Accepting the editor’s suggestion, we have rewritten the abstract as follows (page 1; line 13-27):

“Recently, the growing release of CeO2 nanoparticles (CeO2 NPs) into sewage systems has attracted great concern. Numbers of researches have extensively explored CeO2 NPs' potential adverse impacts on wastewater treatment plants, however, the impaired activated sludge recovery potentials have seldom been addressed yet. To explore physicochemical and biological effects on activated sludge performance and activity recovery of damaged sludge by exposing to CeO2 NPs in sequencing batch reactors (SBRs), four reactors and multiple indicators including water quality, key enzymes, microbial metabolites, microbial community structure and toxicity, were used. Results showed that 10-week exposure to higher CeO2 NPs concentration (1, 10 mg/L) resulted in a sharp decrease in nitrogen and phosphorus removal efficiencies, which were consistent with the tendencies of key enzymes. Meanwhile, CeO2 NPs at concentrations of 0.1, 1, 10 mg/L decreased the secretion of tightly bound extracellular polymeric substances to 0.13%, 3.14%, 28.60%, respectively compared to the control. In addition, 2-week recovery period assays revealed the functional bacteria Proteobacteria, Nitrospirae and Planctomycetes recovered slightly at the phyla level analyzed through high-throughput sequencing, which were consistent with little improvement of effluent performance of the system. It reflected little possibility of activity recovery of damaged sludge.”

Point 2: Lines 54-56. What is the innovation? What is the difference between present and previous research?

Response 2: Thanks for your kind and suggestive comments. Recent researches were only limited to the system performance on the presence of CeO2 NPs exposure. Simultaneously, little is known on how the activated sludge will perform when freed from the chronic exposure of CeO2 nanoparticles and what response microbial community structure will make, which is extremely vital to the stability of the activated sludge system. So the innovation of this paper is to discuss the effect of CeO2 NPs on activated sludge from intermediate exposure period and exposure relief period, and to make a preliminary study on the potential recoverability of CeO2 NPs impaired microorganisms in activated sludge system.

Point 3: The recent study on microbial recovery under NPs stress should be introduced in introduction.

Response 3: Thanks for your kind and suggestive comments. Accepting the editor’s suggestion, we have added some recent study on microbial recovery under NPs stress into our introduction to makes our research background more comprehensive. The added parts of introduction highlighted in italics is shown as follows (page 2; line 75-80).

 “The potential recoverability of CeO2 NPs damaged sludge in nitrogen and phosphorus removal system has seldom been addressed yet. Just limited researches indicated the metabolic recovery capacities of impaired microorganisms under nanoparticles exposure. Recently, only Wu et al. [19] discussed the possibility of microbial activity recovery under ZnO NPs stress when exploring effects of dissolved oxygen to Nitrosomonas europaea.

Also, a reference has been added into the revised manuscript (page 14; line 569-571):

“Wu, J.; Chang, Y.; Gao, H.; Liang, G.; Yu, R.; Ding, Z., Responses and recovery assessment of continuously cultured Nitrosomonas europaea under chronic ZnO nanoparticle stress: Effects of dissolved oxygen. Chemosphere 2018, 195, 693-701,doi.org/10.1016/j.chemosphere.2017.12.078.”

Point 4: The details of the experiment procedures should be stated.

Response 4: Thanks for your kind and suggestive comments. Accepting the editor’s suggestion, we have rewritten the Materials and Methods in the revised manuscript as follows (page 2; line 88-156):

“2.1         Experimental set-up

The activated sludge (AS) was cultured in four SBRs of a working volume 4 L with a concentration of 3.0 g/L biomass. AS was taken from the secondary sedimentation tank at Jiangning Development Zone Sewage Treatment Plant in Nanjing, China. The SBRs were operated daily in 8-h cycles, including a 10-min feeding phase, 2-h anaerobic phase, 4-h aerobic phase, 1-h setting phase, 10-min draining phase, 40-min idle phase. The reactors were fed with synthetic water, which was composed of (mg/L): COD 300, NH4Cl 120, KH2PO4 24, MgSO4·7H2O 20 and CaCl2·5H2O. Micronutrients compositions was shown as following: H3BO3 0.1 g/L, FeSO4·7H2O 6 g/L, CuSO4·5H2O 0.08 g/L, MnCl2·4H2O 0.08 g/L, Na2MoO4·2H2O 0.06 g/L, ZnSO4·7H2O 0.12g /L, CoCl2·6H2O 0.1g/L and NiCl2·6H2O 0.1 g/L. Each reactor was bubbled with volume exchange ratio of 50% through a fine-bubble aerator connected to their bottom. The parameters of the reactor were controlled as following: dissolved oxygen (DO) concentration of 2 mg/L, temperature of 20 ± 2 ℃, and pH value of 7.0-8.0. Hydraulic retention time (HRT) and sludge retention time (SRT) were 8 h and 14 d, respectively.

2.2          CeO2 NPs treatments

The CeO2 NPs (100 mg), purchased from Shanghai Jingchun Scientifical Co. Ltd, first placed in ultrapure water (1.0 L, pH 6.9) for the ultrasonic treatment (25 ℃, 1 h, 250 W, 40 kHz). The mean particle size of CeO2 NPs suspension was 70-150 nm after the analysis by the dynamic light scattering (Malvern Instruments Co. Ltd., UK). 0.1 mg/L CeO2 NPs was chosen as the environmentally relevant concentration, 10 mg/L as the upper concentration limit in the article and 1 mg/L was chosen as the transition value.

2.3          CeO2 NPs exposure and recovery experiment

0.1 mg/L CeO2 NPs was chosen as the environmentally relevant concentration, 10 mg/L as the upper concentration limit in the article and 1 mg/L was chosen as the transition value.

A whole running process was composed of start-up period, intermediate exposure to CeO2 NPs period, and exposure relief period. The SBRs operated continuously for 12 weeks after the system remained stable, including 10-week with CeO2 NPs exposure to study the effect of NPs on activity of active sludge, and 2-week without CeO2 NPs exposure to study the recovery ability of damaged sludge in the first 10 weeks. After 10-week exposure period, the activated sludge in SBRs was washed out three times with deionized water to remove the residual CeO2 NPs in the reactors, and then a 2-week recovery experiment was carried out. At the same time, the total CeO2 and dissolved CeO2 contents were monitored to ensure the NP removal efficiency.

2.4          CeO2 NPs exposure and recovery experiment

During CeO2 nanoparticles loading process, the wastewater effluent samples of the SBRs were collected for ammonia nitrogen (NH4+-N), nitrate nitrogen (NO3--N), nitrite nitrogen (NO2--N), total phosphorus (TP) and chemical oxygen demand (COD) analysis. NH4+-N, NO3--N, NO2--N, TP, biomass concentration and COD were measured in duplicate by Standard Methods [20].

Furthermore, the morphology of the activated sludge was observed by scanning electron microscope SEM (JEOLJEM-1400, 120kV, JAPAN). The soluble microbial products (SMP), loosely bound extracellular polymeric substances (LB-EPS), and tightly bound extracellular polymeric substances (TB-EPS) were extracted separately, as well as the polysaccharides, proteins, and humic acids content by Hou et al. [21] and Wang et al. [15]. The lactate dehydrogenase (LDH) release assay was measured with a LDH kit (Jiancheng Bioengineering Co. Ltd., Nanjing, China) [11]. And detection of the microbial diversity could be carried out by the high-throughput sequencing technology for microbial analysis. In addition, measurements of ammonia monooxygenase (AMO), nitric oxide reductase (NOR), nitrate reductase (NAR) and nitrite reductase (NIR) activity were performed according to the method of Zheng et al. [22]. Microbial community structure was analyzed by miSeq high-throughput sequencing according to the method of Li et al. [23]. Triplicate samples were collected from each SBR reactor to ensure the integrity of the AS samples, and all results were presented on average.

2.5          Energy dispersive spectroscopy analysis

The mixed solution (40 mL) was firstly dispersed into four centrifuge tubes (each 10 mL). These four centrifuge tubes were then centrifuged at 3,000 rpm/min for 5 min. After pouring the supernatant, the lotion (1.0 mmol/L EDTA, 0.1 mol/L NaCl) was added to the centrifuge tubes to restore the original volume (10 mL). Then the mixtures were transferred to the oscillating tube (50 mL) and oscillated in a constant temperature oscillator at 150 rpm/min for 30 min to remove CeO2 NPs adsorbed on the surface of the sludge [24]. After repeating the steps above, the centrifuge tubes were centrifuged at 3,000 rpm/min for 5 min again. Then pour out the supernatant and dry the remaining sludge at 60 ℃. Finally, the dried sludge was ground into powder for the spectrum analysis, which could be used to determine the proportion of intracellular cerium.

2.6          Data analysis

All assays were conducted in triplicate. For statistical analysis, the experimental values were compared to their corresponding control values. An analysis of variance (ANOVA) was used to test the significance of the results and p < 0.05 was considered to be statistically significant, as similarly described by Cao et al. [25].”

Also, a reference has been added into the revised manuscript (page 14; line 581-583):

“Li, Y.; Zhang, J.; Zhang, J.; Xu, W.; Mou, Z., Microbial Community Structure in the Sediments and Its Relation to Environmental Factors in Eutrophicated Sancha Lake. 2019, 16, (11), 1931,doi.org/10.3390/ijerph16111931.”

Point 5: Why the nitrite accumulation reduced under 10 mg/L CeO2 NPs? Why is there no change in AMO activity, but the ammoxidation inhibition is detected?

Response 5: Thanks for your kind and suggestive comments. Nitrite mainly accumulated in the aerobic stage, and the main source of nitrite was the conversion of NH4+-N to NO2--N under the action of AMO.

During the 10-week exposure period, the inhibition of CeO2 NPs on activated sludge

was in positive correlation with the CeO2 NPs exposure concentration. In Figure. 4 (a) (page 7; line 267), it can be seen that the relative activity of AMO under chronic 10 mg/L CeO2 NPs stress was lower than that under short-term exposure. Although the relative activity of AMO decreased less than that of NIR and NAR, the reduction of nitrite accumulation in aerobic phase also confirmed the inhibitory effect of CeO2 NPs on AMO.

Point 6: Lines 206-315. Should be further deleted and highlighted. How can high-throughput sequencing data determine the accuracy of the results without setting duplicates?

Response 6: Thanks for your kind and suggestive comments. The high-throughput sequencing data discussed in the article were represented by mean values, which have been explicitly proposed in the rewritten Materials and Methods. The details are as follows (page 3; line 138-141):

“Microbial community structure was analyzed by miSeq high-throughput sequencing according to the method of Li et al. [23]. Triplicate samples were collected from each SBR reactor to ensure the integrity of the AS samples, and all results were presented on average.”

Also, a reference has been added into the revised manuscript (page 14; line 581-583):

“Li, Y.; Zhang, J.; Zhang, J.; Xu, W.; Mou, Z., Microbial Community Structure in the Sediments and Its Relation to Environmental Factors in Eutrophicated Sancha Lake. 2019, 16, (11), 1931,doi.org/10.3390/ijerph16111931.”

Point 7: Why is there only high-throughput sequencing data in the recovery phase without regular water quality indicators?

Response 7: Thanks for your kind and suggestive comments. During the two-week recovery period, we investigated the stability of the damaged sludge system and the shift of microbial communities. In the system stability assessment, we carried out water quality monitoring and energy spectrum analysis of damaged sludge. The water quality is detailed as follows (page 10 line 378-382):

“It was observed that the performance of activated sludge in the reactors did not recover significantly within 2-week recovery time relieved from the 10-week exposure to different CeO2 NPs concentration. In R4, the removal efficiencies of nitrogen and phosphorus were stable at 86.12% and 45.26%, respectively, which were only 1.09% and 4.63% higher than those during the period of exposure.”

Point 8: There is no point in discussing the microbial diversity based on phyla level.

Response 8: Thanks for your kind and suggestive comments. To further explore the shift of microbial community in damaged sludge during the recovery period of 2 weeks, the shift of microbial communities at the phyla level in four reactors were discussed. There were Nitrospirae, Acidobacteria, Chloroflexi and other common bacteria in SBRs, which all played an important role in sewage treatment. At the same time, There were numbers of functional bacteria which were closely related to nitrogen and phosphorus removal. Exploring their changes was helpful to discuss the mechanism of nitrogen and phosphorus removal in depth, so it is necessary to discuss the microbial diversity based on phyla level.

Point 9: Lines 397-400. The past literature has already systematically reported the whole process of SBR being under CeO2 stress. What is your innovation? 

Response 9: Thanks for your kind and suggestive comments. The whole process of this manuscript refers to the performance of activated sludge in the 10-week CeO2 NPs exposure period and 2-week exposure relief period. Previous studies only focused on the effect of CeO2 NPs to activated sludge during the intermediate exposure period. Few researchers discussed the performance of sludge recovery during exposure relief period. This is also the innovation of this article, which is different from previous work. At the same time, the framework of this paper can be clearly understood by the following graphical abstract.

Please see the attachment for detailed information.

Reviewer 3 Report

The study is interesting but, in my opinion, it has a limited relation with real cases.

The authors mentioned to the release of CeO2 NPs into sewage collection systems and the consequent problems to WWTPs. I imagine that these releases are only due to industrial activities and CeO2 NPs are not commonly found in WW of exclusively human origin. Furthermore, I imagine that if an industrial WW contains CeO2 NPs, it is pre-treated with dedicated pre-treatment before being discharged into the sewage collection systems. Finally, which is a realistic concentration of CeO2 NPs at the entrance of a WWTP? This point must be treated more in detail. In my opinion the study considered too high, non-realistic concentration values.

Author Response

Response to Reviewer 3 Comments

Point 1: The study is interesting but, in my opinion, it has a limited relation with real cases. The authors mentioned to the release of CeO2 NPs into sewage collection systems and the consequent problems to WWTPs. I imagine that these releases are only due to industrial activities and CeO2 NPs are not commonly found in WW of exclusively human origin. Furthermore, I imagine that if an industrial WW contains CeO2 NPs, it is pre-treated with dedicated pre-treatment before being discharged into the sewage collection systems. Finally, which is a realistic concentration of CeO2 NPs at the entrance of a WWTP? This point must be treated more in detail. In my opinion the study considered too high, non-realistic concentration values.

Response 1: Thank you very much for kindly comments on the manuscript. We are sincerely grateful to your review. In this article, we set up three concentration gradients: 0.1 mg/L, 1 mg/L, 10 mg/L and blank control group. 0.1 mg/L CeO2 NPs was chosen as the environmentally relevant concentration in wastewater systems due to the following article (Nano-CeO2 exhibits adverse effects at environmental relevant concentrations.doi: 10.1021/es103309n). Meanwhile, in view of the rapid applications and disposal of CeO2 NPs products, the release of CeO2 NPs in the environment is increasing, and acute leakage may be inevitable, thus a concentration of 10 mg/L CeO2 NPs was considered. (Responses of wastewater biofilms to chronic CeO2 nanoparticles exposure: Structural, physicochemical and microbial properties and potential mechanism; doi: https://doi.org/10.1016/j.watres.2018.01.031). 1 mg/L was chosen as the transition value between 0.1 mg/L and 10 mg/L. In recent studies, the concentration distribution of nano-materials in water environment is limited, but some studies showed that about 8 300 metric tons of nanomaterials have released into the environment through diffusion and other behaviors until 2014 (Predicted Releases of Engineered Nanomaterials: From Global to Regional to Local. Doi: 10.1021/ez400106t). Considering the rapid development and consumption, the number of nanomaterials in the environment is probably more. In a word, this study is based on the realistic situation of theoretical research, has a certain practical basis, and has fully taken into account the extreme situation of a large number of nano-materials leaking into the water body.

Please see the attachment for detailed information.

Round 2

Reviewer 2 Report

The author made some revision of the manuscript, the content of the article was clearly stated. However, there are still some points that the author needs to seriously answer and modify:

Point 1: How to add the expected concentration of NPs into the reactor at the beginning of each cycle.

Point 2: How to ensure the uniform concentration of NPs entering the reactor per cycle.

Point 3: 2.3 “the activated sludge in SBRs was washed out three times with deionized water to remove the residual CeO2 NPs in the reactors”

Unlike pure bacteria, NPs were wrapped in sludge flocs or trapped in EPS after stress, which was difficult to removal. How could you ensure that NPs were eliminated before recovery experiment?

Point 4: 2.3 “At the same time, the total Ce and dissolved Ce contents were monitored to ensure the NP removal efficiency”

I don't see any data on the distribution of Ce in the reactor during the stress and recovery period.

Point 5: “Response 5: Thanks for your kind and suggestive comments. Nitrite mainly accumulated in the aerobic stage, and the main source of nitrite was the conversion of NH4+-N to NO2--N under the action of AMO.”

Decreased nitrite accumulation might also be the result of inhibition of nitrate reduction, which should also be discussed.

Point 6: Table 1. The Alpha diversity of the four activated sludge reactors.

Put all the sample results (Including repetition) in the table.

Point 7: In the high-throughput sequencing section, the author should focus on the genus level rather than the phyla level.

Point 8: The abundance change of microorganisms at the genus level should be placed in the manuscript.

Reviewer 3 Report

in the new version of the manuscript the authors did not provide substantial changes to justify a change in my final decision. Moreover, they introduced a new section (2.3) with the same title of a previous section (2.4 in the new version). This generates some confusion.